# A rare oasis effect for forage fauna in oceanic eddies at the global scale

Aurore Receveur [1,2] ✉, Christophe Menkes [3], Matthieu Lengaigne [4], Alejandro Ariza[4,5], Arnaud Bertrand [4], Cyril Dutheil [4,6], Sophie Cravatte [7,8], Valérie Allain [1], Laure Barbin [1,3], Anne Lebourges-Dhaussy [9], Patrick Lehodey [1,10] & Simon Nicol[1,11]

Oceanic eddies are recognized as pivotal components in marine ecosystems, believed to concentrate a wide range of marine life spanning from phytoplankton to top predators. Previous studies have posited that marine predators are drawn to these eddies due to an aggregation of their forage fauna. In this study, we examine the response of forage fauna, detected by shipboard acoustics, across a broad sample of a thousand eddies across the world's oceans. While our findings show an impact of eddies on surface temperatures and phytoplankton in most cases, they reveal that only a minority (13%) exhibit significant effects on forage fauna, with only 6% demonstrating an oasis effect. We also show that an oasis effect can occur both in anticyclonic and cyclonic eddies, and that the few high-impact eddies are marked by high eddy amplitude and strong water-mass-trapping. Our study underscores the nuanced and complex nature of the aggregating role of oceanic eddies, highlighting the need for further research to elucidate how these structures attract marine predators.

Oceanic mesoscale eddies are coherent and transient swirling structures, ubiquitous in the world's oceans[1], and often considered oceanic oases that aggregate marine life in pelagic deserts[2–5]. Evidence from fishing catch[5,6] and satellite-tracking data[7–9] reveals that a preference for Anticyclonic Eddies (AE) for hosting a variety of marine predators, including fish, sharks and turtles. Two recent studies demonstrated higher abundance of predatory fish (e.g. tuna) and fishing activity in AE across regional and global spatial scales respectively[5,6]. The prevailing explanation for this aggregation is a bottom-up structuring effect, wherein increased food resources within AE attracts predators. These predators primarily feed on forage fauna, encompassing a wide variety of small fishes, crustaceans and molluscs[10,11]. While some studies have

observed similar forage fauna aggregation in AE[2,12,13], other have revealed a more nuanced response, with either aggregation in Cyclonic Eddies (CE)[14,15], no discernible effect[16], or a variable signal highly dependent on the characteristics of the eddy, such as age, size, amplitude or lifespan[16–18]. Understanding the processes governing the aggregation of forage fauna in mesoscale eddies remains a complex endeavour. The prevailing hypotheses include the bottom-up structuring effect (Table S1), positing increased chlorophyll within CE as a lure for zooplankton, consequently boosting forage fauna[14,19,20]. Alternatively, a trapping effect is proposed, wherein a micronektonic community becomes physically trapped within eddies, evolving differently from surrounding communities as the eddy moves[18]. A less

[1]Oceanic Fisheries Programme, The Pacific Community, BP D5 98848 Noumea, New Caledonia. [2]CESAB, FRB; 5 Rue de l'École de Médecine, 34000 Montpellier, France. [3]ENTROPIE, IRD, CNRS, Ifremer, Université de la Réunion, Université de la Nouvelle-Calédonie, Nouméa, New Caledonia. [4]MARBEC, Université Montpellier, IRD, Ifremer, CNRS, Sète, France. [5]DECOD, Ifremer, INRAE, Institut Agro, Nantes, France. [6]Department of Physical Oceanography and Instrumentation, Leibniz Institute for Baltic Sea Research Warnemünde, Rostock, Germany. [7]Université de Toulouse, LEGOS (IRD, CNES, CNRS, UT3), Toulouse, France. [8]IRD, Noumea, New Caledonia. [9]LEMAR, IRD, Univ. Brest, CNRS, Ifremer; BP70, 29280 Plouzané, France. [10]Mercator Ocean international, 31400 Toulouse, France. [11]Institute for Applied Ecology, Centre for Conservation Ecology and Genomics, University of Canberra, Bruce ACT 2617, Australia. ✉e-mail: aurore.receveur@9online.fr

prevalent hypothesis suggests the physical convergence of water masses in AEs[2,21], facilitating the aggregation of micronekton organisms.

However, a comprehensive assessment of how forage fauna respond to eddies is currently severely hampered by the scarcity of observations. While satellite technology enables the remote sensing of surface temperature and phytoplankton responses[22,23], studies on marine predator often rely on fishing or satellite tracking data[5,8]. Yet, observing forage fauna communities in the oceanic food web data remains challenging because direct collection of these communities typically requires oceanographic cruises equipped with net tows or ship-borne acoustic echosounders capable of continuously scanning sound-scattering ocean fauna down to -1000 m[24,25]. While net tows are effective in species examination, they struggler to adequately resolve mesoscale structures without dense, repeated sampling across multiple depth ranges and surveys. On the other hand, although acoustic echosounder data lack the ability to identify the species without specific validation, they offer high-resolution insights into the vertical structure of forage fauna when crossing eddies[15,17].

To date, the exploration of the eddy-enhanced forage biomass hypothesis in marine predator studies has predominantly relied on two seminal publications[2,13]. These studies, analysing the ship-borne acoustic response in 13 and 4 eddies, respectively, concluded an oasis eddy effect in AE. However, the limited sample size in these and other related studies (Table S1) hinder a thorough assessment of the eddies' effect on forage fauna.

The main objective of this study is to systematically evaluate the influence of eddies on forage fauna and to scrutinize the eddy-enhanced forage biomass hypothesis. To achieve this, we merged an extensive database of acoustic vertical profiles spanning the upper 750 m of the ocean[26] within a global eddy database spanning the 2001–2020 period[27]. This comprehensive approach allowed us to examine the effect of 999 eddies on forage fauna consistently across diverse oceanic regions. By extracting key eddy characteristics (size, amplitude, lifespan, trapping ability, temperature and chlorophyll signatures), we aim to infer their influence on forage fauna. While our findings do indicate an eddy-induced effect on sea surface ocean temperature (SST) and surface chlorophyll concentration, with opposite effects observed between CE and AE in the majority of cases, it also unveils that only exceptionally strong AE and CE are capable of aggregating forage fauna, suggesting that the oasis effect of eddies is likely more an exception than the rule.

## Results and discussion

We used publicly available sonar data (Table S2, with the exception of the three Mozambique channel surveys) derived from acoustic surveys conducted over various regions of the global ocean. The water column backscatter data at 38 kHz was vertically integrated into to derive the Nautical Area Scattering Coefficient (NASC, $m^{-2}$ $nmi^{-2}$), which serves as an indicator of forage fauna biomass[28]. It is noteworthy that the 38 kHz frequency only detects a fraction of the total forage fauna community, primarily gas-filled organisms (e.g. fish with gas-filled swimbladders and siphonophores)[29,30], potentially overlooking other components of the community.

This acoustic database covers >350,000 km around the globe, spanning depth from 20 to 750 m and the time frame from 2001 to 2020[26] (Fig. S1). The spatial resolution of this acoustic data varies, being available either at 1-km or 1-nautical mile intervals, contingent upon the survey conducted. We collocated this dataset with a global eddy database derived from satellite altimetry[27], enabling the sampling of acoustic signature in 999 eddies (Fig. 1A), evenly distributed between AE and CE (473 and 526 eddies, respectively, Fig. 1B).

For each sampled eddy, the acoustic observations (i.e. the NASC vertical profiles, Fig. 1C) were categorized as 'Inside' and 'Outside' of the eddy, based on the exact detected eddy contour (see Methods and

Fig. 1D). The 'Outside' region was defined as the spatial ribbon outside the eddy, extending from the effective border to twice the eddy radius. To calculate acoustic anomaly values for each sample eddy, we compare the inside and outside profiles using the formula: anomaly = (mean of inside profiles − mean of outside profiles)/(mean of outside profiles). These anomalies were computed exclusively during coherent day-time and night-time periods to mitigate the potential blurring effect caused by diel vertical migration (e.g. the inside profile sampled during the day was compared to the outside profile also sampled during the day, and vice-versa for the night-time).

To verify the inclusion of the widely recognized SST and surface chlorophyll-a concentration (Chl) signatures of eddies in our dataset, we additionally retrieved remotely-sensed SST at 5 km spatial resolution[31] and Chl at 4 km spatial resolution[32]. These data points were extracted at the nearest spatial and temporal locations of our collocated eddy/acoustic database.

### A subdued forage fauna response in contrast to clear SST and Chlorophyll signals

It is commonly accepted that CE typically exhibit significantly colder and chlorophyll-enriched surface signals within their core, while AE display opposite signals[33–35]. Our dataset confirms these expectations, revealing a significant surface cooling of ~−0.1 °C within CE compared to surrounding waters ($p$-value $< 2E^{-16}$) while AE exhibits a milder yet still significant surface warming of +0.05 °C ($p$-value $= 4E^{-5}$) (Fig. 2A). These observed signs and magnitudes of these are in line with previous global assessments[34,36]. Similarly, an analysis performed on Chl indicates that CE are associated with a significant albeit modest Chl increase by 3.2% on average ($p$-value $= 2E^{-3}$), while the average Chl anomalies within AE are marginally significant ($p$-value $= 0.051$) (Fig. 2B).

However, contrasting with the SST and Chl responses, our analysis does not uncover a mean NASC signature in the upper 0–200 m layer for either eddy type. Instead, there is only a slight -5% NASC increase at depth for AE, and a corresponding -2% decrease for CE (Fig. 2C). Further examination of the vertical structure of the acoustic profiles indicates that the mean NASC profile inside AE does not significantly differ from that outside of the eddy (Fig. 2D). In contrast, within CE, a significant mean NASC decrease of about 10% is detected between 450 and 600 m. Subdividing these findings based on distinct eddy regions (e.g. the core, the border) did not yield significant differences (Fig. S2)[21,37].

To further detail the effects of eddies, we assess the percentage of eddies where the signals of SST, Chl, averaged NASC within 0–200 m or averaged NASC within 200–750 m inside the eddy was not significantly different to those outside (termed "Null" effect), significantly higher than the outside (termed "Increase") or significantly lower than the outside (termed "Decrease") at 95% confidence level (See Methods and Fig. S3). The majority of eddies show a significant response in both SST and Chl (87% for SST and 82% for Chl; Fig. 3). Among the CE exhibiting a significant SST response, two-thirds experience a cooling of -0.5 °C, while the remaining third display a warming of -0.2 °C. Similarly, the majority (-60%) of AE are warmer by -0.3 °C than their surroundings, consistent with previous literature[34,35]. However, about 40% of AE cores register colder temperatures than their surroundings by -0.2 °C. This diversity is consistent with recent findings from studies analysing regional and global SST satellite data, indicating that surface cold-core AE and warm-core CE are also abundant in the global ocean, accounting for 20–45% of the eddies, depending on the region considered[36,38,39]. Regarding Chl, the sign of the response is even more heterogeneous, with half of CE and AE showing -18% of richer waters relative to their surroundings, while the other half indicate 12 of poorer waters inside (Fig. 3). This diversity has been attributed to the diversity of mechanisms operating in different regions, with diverse impacts, such as Chl horizontal advection ("stirring") around the eddies'

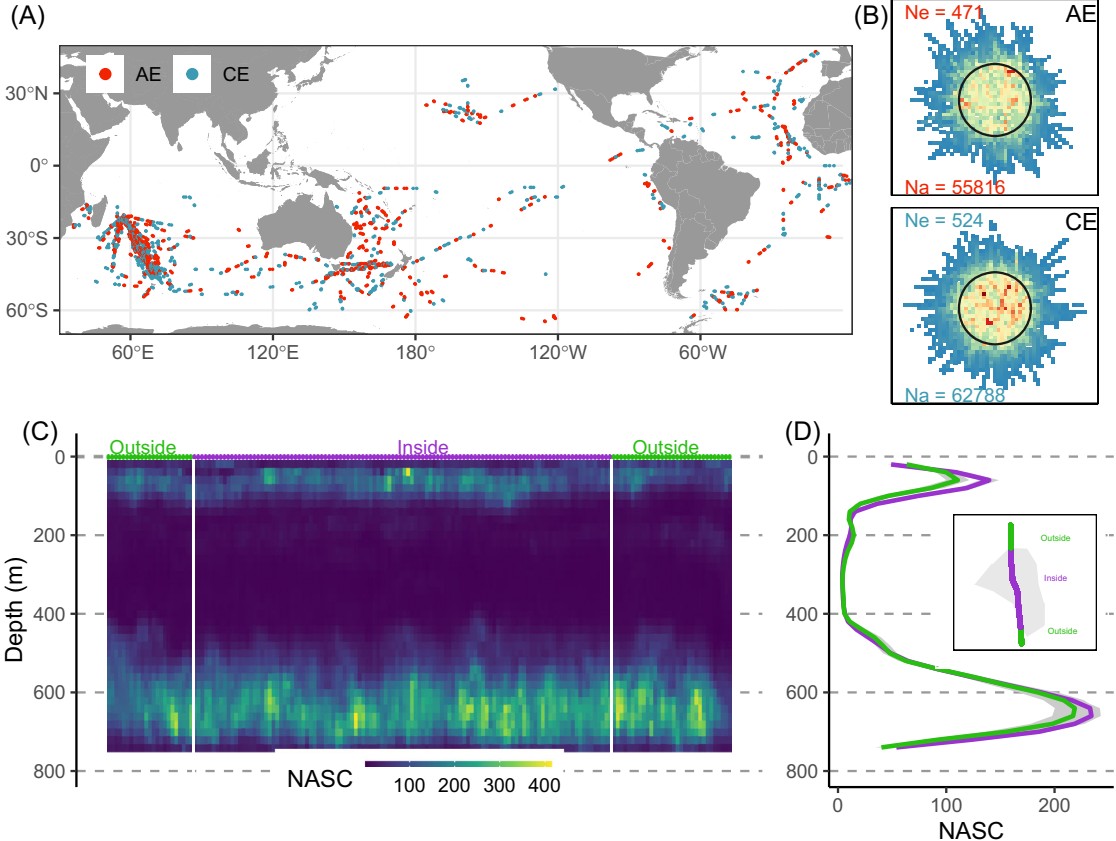

**Fig. 1 | Overview of the collocated datasets and one example of eddy sampled by acoustic. A** map of acoustic data sampling cyclonic (CE, *blue*) and anti-cyclonic (AE, *red*) eddies; (**B**) density of acoustic tracks around and within eddies (from blue to red, dark cycles represent the normalized effective radius) for anti-cyclonic (AE, *top panel*) and cyclonic (CE, *bottom panel*) eddies; numbers indicate the number of sampled eddies ("*Ne =*", *top*) and the number of acoustic profiles ("*Na =*", *bottom*) in each eddy type; (**C**) Example of one acoustic track section in an anticyclonic eddy showing scattering sound layers of marine fauna (unit: Nautical Area Scattering Coefficient, NASC in $m^2$ $mn^{-2}$). The vertical white bars denote the separation between the "inside" and "outside" of the sampled eddy, as depicted in (**D**) by the grey polygon and the acoustic track from the ship. In (**D**), the vertical mean NASC profiles in the inside (purple) and in the outside (green) areas are drawn together with the 95 % confidence interval (1.96*standard deviation/root square of number of observations) of the mean values. Source data are provided as a Source Data file.

peripheries, phytoplankton transport within eddy cores, upwelling and downwelling effects, as well as eddy-induced changes in stratification[23].

Our investigation revealed significant anomalies in both positive and negative directions for SST and chlorophyll both in CE and AE, consistent with findings in previous literature[34–36,38,39], thus confirming the robustness of our methodology. Conversely, the examination of forage fauna reveals a notable absence of significant changes (Fig. 3) in most CE and AE (86% and 90%, respectively, showed no effect in the epipelagic layer, and in 84% and 88%, respectively, in the mesopelagic layer). In the epipelagic layer, a significant increase (i.e. an oasis effect) in forage fauna is found in only 5% and 7% of AE and CE, respectively. The most pronounced indication of an oasis effect is observed for AE in the mesopelagic layer, where 9% of eddies exhibit a significant increase in forage fauna, whereas CE show a slightly larger proportion (11%) of forage fauna decrease at these depths. Even when focusing on the mesopelagic layer between 400–600 m, where a significant signal is evident in Fig. 2D for CE, the percentage of eddies exhibiting no effect marginally decreases from 84% to 76% (Fig. S4).

The influence of eddies is expected to penetrate deeply into the water column, particularly for the strongest eddies[35]. Given that a large proportion of mesopelagic species engages in diel vertical migration, wherein they remain at the surface during the night and dive at depth during the day[40]–there is a potential scenario where these mesopelagic organisms evade the eddy influence by swimming to deeper levels. However, our acoustic data is limited by the range of depths covered by our acoustic data, which only extends up to 750 m. Therefore, we are unable to directly test this hypothesis using our dataset.

## Processes at play in forage fauna response

To broaden our analysis, we assess the dependence of the NASC response on six key eddy characteristics, which have been identified in previous studies as potential contributors to the forage fauna response within eddies (Table S1)[2,13,15,17,41]: amplitude, SST anomaly, Chl anomaly, trapping metric, effective area and lifespan. Upon comparing eddies between increasing, decreasing and null-effect, we find significant differences in only two of these characteristics: amplitude and eddy trapping capacity. Stronger eddies and those with higher trapping capacity exhibit a greater tendency to both increase and decrease the density of forage fauna (Fig. 4). Eddy amplitudes serve as proxies for both the rotational current strength and the vertical density structure shape, with higher amplitude associated with deeper isotherms in AE and negative amplitude associated with shallower isotherms in CE. Given that high amplitude was associated with both increased and decreased forage fauna density (Fig. 4), we posit that the amplitude effect on forage fauna primarily reflects a physical barrier effect, where the horizontal current strength creates a barrier separating the inner and outer water masses, rather than an effect due to the vertical isothermal displacement. Furthermore, our results highlight a significant effect of eddy trapping capacity, which aligns with the trapping water mass effect previously suggested in the literature[1,18].

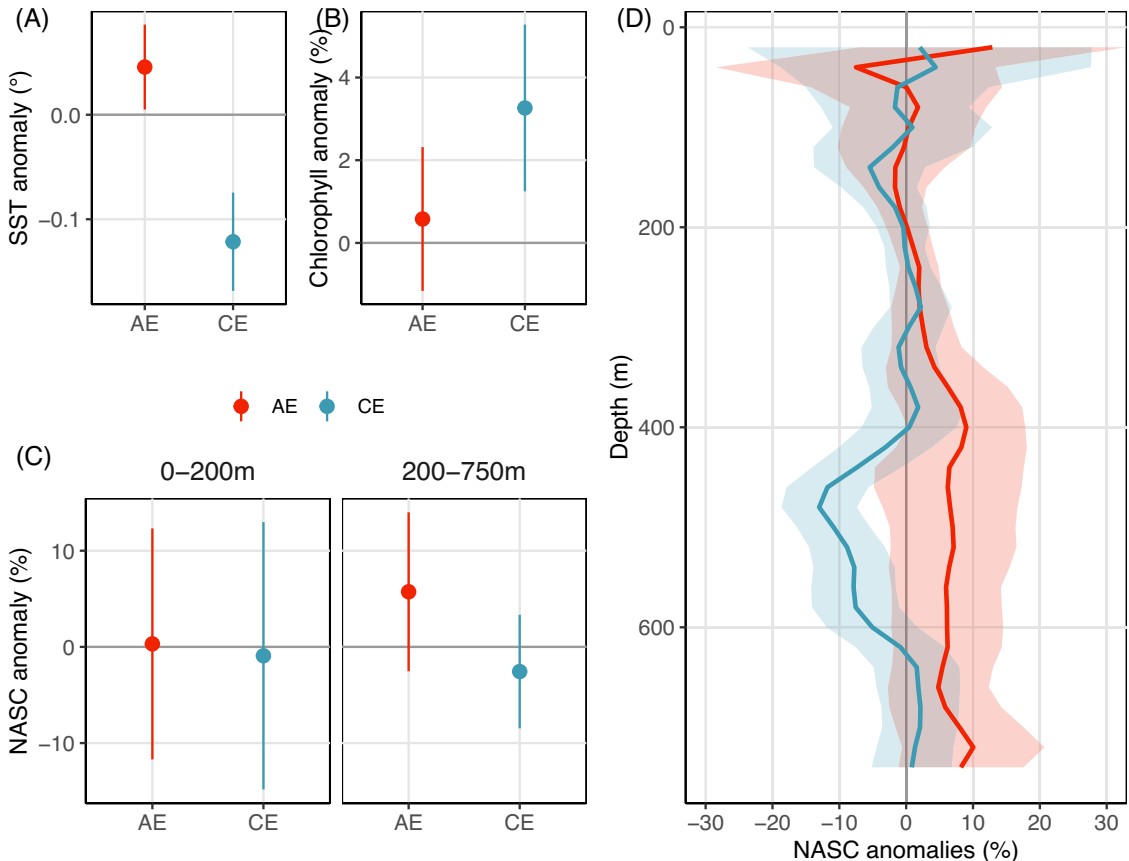

**Fig. 2 | Mean temperature, chlorophyll and acoustic responses in Cyclonic and Anticyclonic eddies.** Mean anomalies across the 956 sampled eddies (i.e. signal changes in % of inside versus outside eddies) for (**A**) SST, (**B**) surface chlorophyll, (**C**) Acoustic backscatter anomalies averaged in 2 vertical layer (epipelagic: 0–200 m and mesopelagic: 200–750 m *rows*) and (**D**) Acoustic backscatter anomalies across depth for AE (red) and CE (blue). Vertical bars (**A**), (**B**) and (**C**) and vertical ribbons (**D**) show the 95 % confidence interval (1.96*standard deviation/ root square of number of observations) of the mean values. Source data are provided as a Source Data file.

Based on our findings, it is probable that the principal mechanism explaining the impact of high-amplitude eddies on forage fauna is the trapping water mass effect observed in both strong AE and CE (Fig. S5). This phenomenon can lead to a biological community within the eddy that differs from its surroundings, potentially leading to increased or decreased biomass within the eddy. In essence, a forage fauna community may be trapped within a strong, trapping eddy and evolve relatively autonomously as the eddy moves, thereby a discernible difference between inside and outside. A previous study has noted an acoustic signature inside an eddy differing from the surrounding waters but similar to the acoustic signal of the region where the eddy originated 27 days earlier[18]. This study proposed that the studied eddy trapped waters within its core, thus preserving the physical and biological signal of the water from its origin 1 month prior. Our results, derived from numerous eddies, concur with these findings. Another explanation posited to explain the forage effect in these few eddies is that strong amplitude eddies can physically trap zooplankton and thus attract forage fauna which is more mobile[15]. However, our data, limited to micronekton, precludes us from assessing this potential zooplankton attraction effect.

Our study revealed no significant difference in SST signal between "null-effect" eddies and those with detected impacts, challenging the hypothesis proposing a warmer niche in AE that fosters a quicker growth of forage fauna organisms[13,41]. Similarly, the absence of effects in Chl signal between forage-affecting and "null-effect" eddies questions the commonly discussed bottom-up effect, where the aggregation of phytoplankton would eventually lead to the aggregation of the

upper trophic food web[2,14,15]. However, our analysis solely relied on surface Chl signals as measured by satellite, overlooking the assessment of the vertical Chl structure altered by eddies, which often involves a deepening of Chl maxima within many AEs[42]. Due to the unavailability of data on this vertical Chl structure, we were unable to explore whether alterations in vertical Chl profiles might influence forage fauna response to eddies.

In strong amplitude eddies, relying solely on the amplitude is inadequate for predicting the eddy's influence on forage fauna, as not all strong eddies significantly affect it. Even when considering the top 20% strongest eddies (amplitude > 0.1 m), the majority (82%) demonstrate no detectable effect, with only 17% of strong AE showing a significant increase in forage fauna at depth (Fig. S6). Further narrowing the focus to the 5% strongest eddies (amplitude > 0.22 m), forage fauna signals remain undetectable for 70% of eddies, with only 35% of AE displaying a significant increase in forage fauna at depth (Fig. S6).

### Regional sensitivity

The overwhelming majority of oceanic eddies identified in our dataset exhibit no discernible impact on forage fauna density, with consistently <10% of eddies exhibiting what is known as the "oasis effect" (i.e. an aggregating effect, 5%, 7%, 9% and 6%; Fig. 3) for both AE and CE in both the epipelagic and mesopelagic layers. A closer examination of the regional response of forage fauna across Longhurst[43] biogeochemical provinces (Fig. S7) reveals that regional results for most sampled provinces align with the global analysis, with an overwhelming proportion of "null-effect" eddies (from 55 to 100% in the

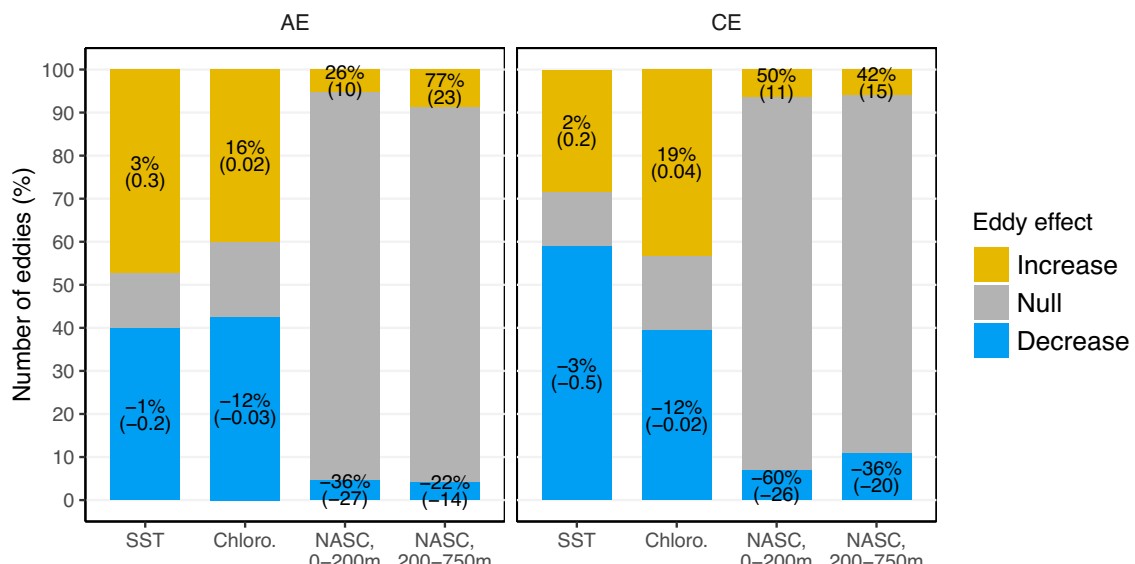

**Fig. 3 | Proportion of affecting eddies on ocean temperature, primary production and biology.** Eddy percentage significantly (based on a Wilcoxon distribution test at 95% confidence level) influencing (*yellow* and *blue*) or not (*grey*) SST (Sea Surface Temperature), chloro (chlorophyll) and forage fauna density (NASC) in two vertical layers (*x-axis*) for AE and CE (*columns*). For each of the four variables, the numbers indicate the mean change values in percentage for eddies with positive NASC anomalies ("increasing", *yellow*) and negative NASC anomalies ("decreasing", *blue*) compared to eddies with no effect ("null", *grey*), and absolute mean values are indicated in brackets below the percentage. Source data are provided as a Source Data file.

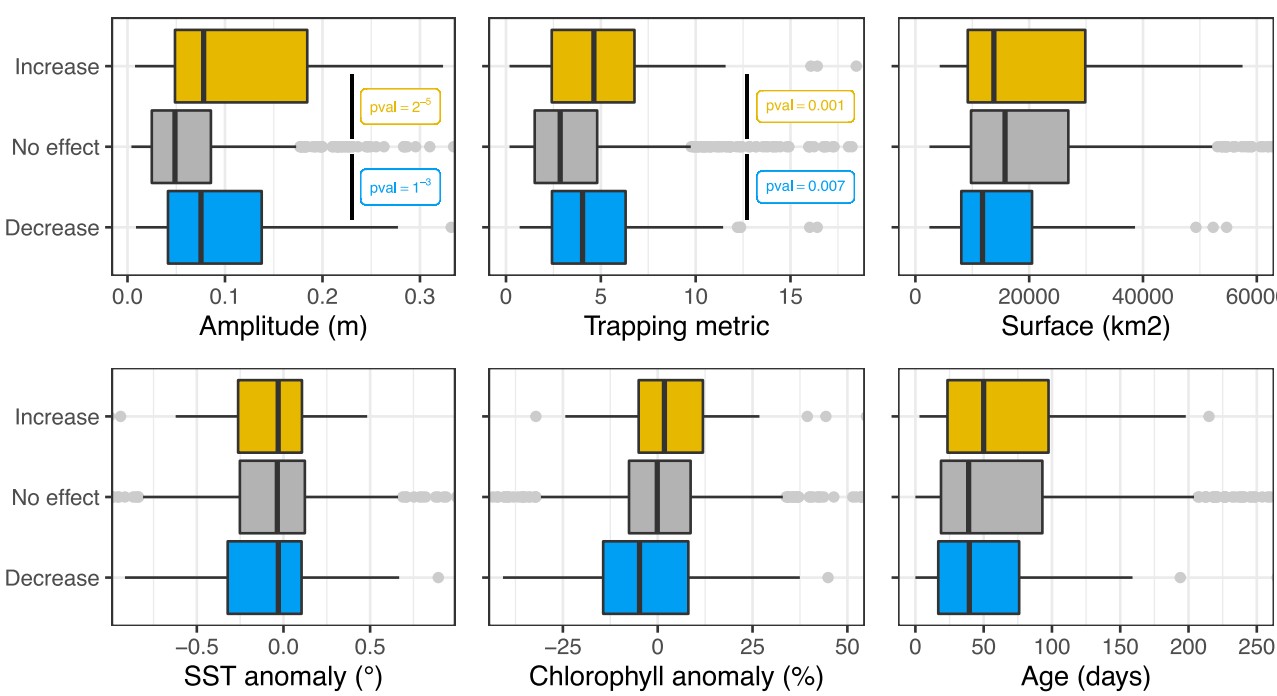

**Fig. 4 | Eddy characteristics of increasing, decreasing and null-effect eddies on forage fauna.** Distributions of the 6 eddy characteristics (amplitude, trapping ability, surface area, SST eddy signal, chlorophyll eddy signal and eddy age) in the eddies (AE and CE pooled) increasing (*yellow*), decreasing (*blue*), and null-effect (*grey*) the forage fauna in the epipelagic layer. *p*-values < 0.001 are written for the characteristics with a significant (two-sided Wilcoxon test with a 5% level) difference between the increasing/decreasing and null-effect distributions. $n = 62$ increasing, $n = 65$ decreasing and $n = 864$ null-effect independent eddies were tested in the Wilcoxon tests. The boxplot bounds are the 0.25 (Q25) and 0.75 (Q75) quantiles, the inside vertical line shows the median, and the horizontal segments are the minimal value [Q25 − 1.5*(Q75-Q25)], and the maximal value [Q3 + 1.5*(Q75-Q25)]. Source data are provided as a Source Data file.

sampled provinces). Nevertheless, three specific provinces - the East African coast, the Sub-Antarctic water ring and the Antarctic−exhibit slightly higher proportions of influencing eddies compared to the global results, although the dominance of null-effect eddies persists

[85 (AE) and 68% (CE); 79 (AE) and 72% (CE); and 66 (AE) and 63% (CE) respectively]. Eddies with significant effects on forage fauna tend indeed to cluster in subtropical regions and southern latitudes, in high energetic current systems such as the South Subtropical Convergence

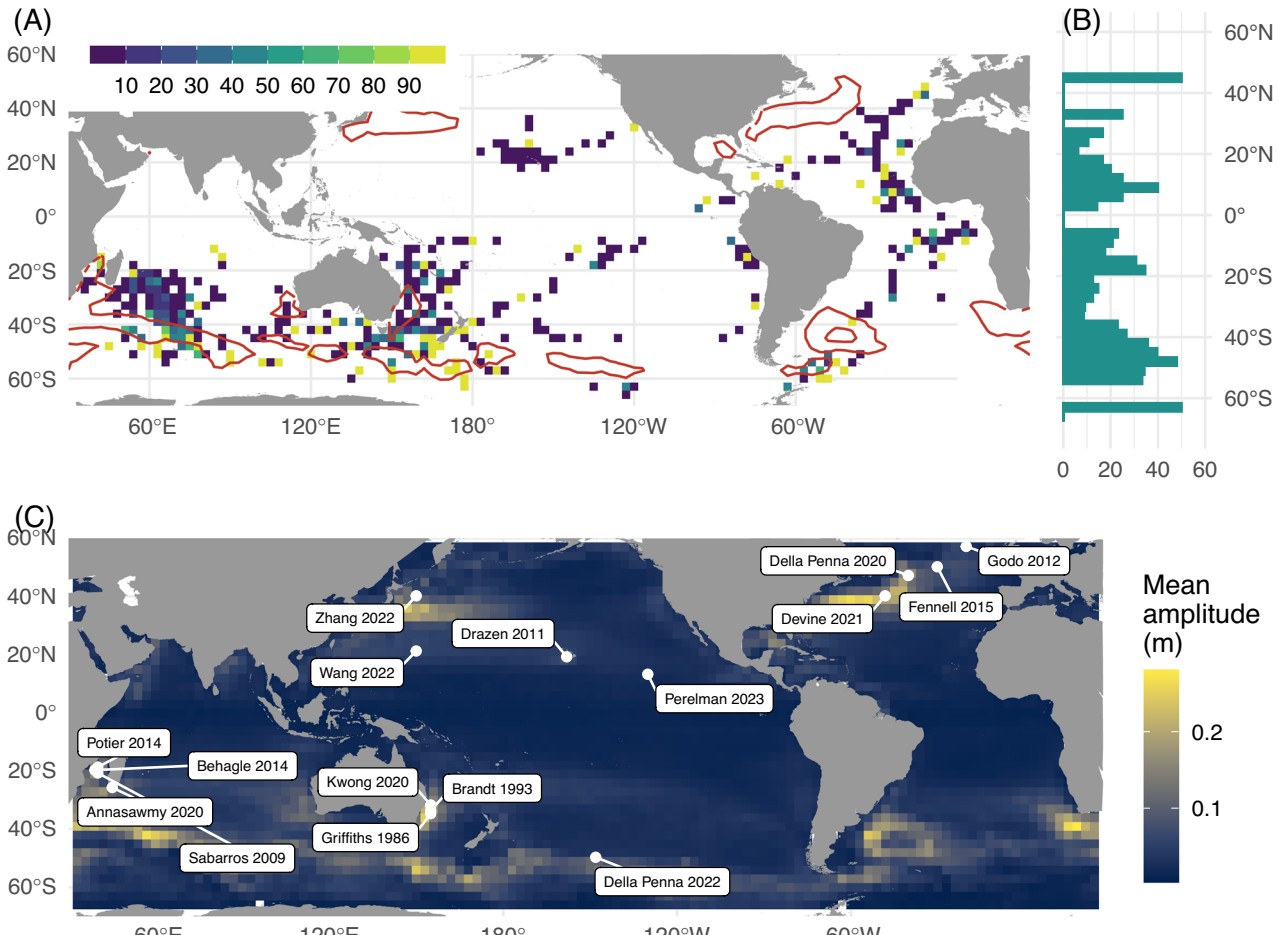

**Fig. 5 | Spatial sensitivity of the results and comparison to previous publications.** Map showing the percentage of eddies having a significant effect (increasing and decreasing combined) on forage fauna among all eddies sampled in one grid cell on a 3° spatial grid (e.g. a grid cell with 10 sampled eddies, including 6 with a significant effect would show 60%) (*colours*), (**B**) and the corresponding latitudinal distribution. **C** Mean amplitude (*colours*) of all eddies detected over the period 1993–2021 on the same 3° spatial grid as (**A**). The crossed squares indicate the mean localisation of previous studies looking the effect of mesoscale eddies on forage fauna (see Table S1 for complete references). The localizations of the published papers were extracted from the different studies, and when the study was spread, the mean study localisation was computed. Red contours in (**A**) represented regions with mean amplitude >0.1 m extracted from (**C**). Source data are provided as a Source Data file.

province[44] and where eddies have strongest mean amplitudes (Fig. 5A, B). Notably, previous studies[13,41,45] highlighting a significant effect of eddies on forage fauna, often reporting an increasing fauna effect in CE, were primarily located in these specific high-amplitude eddy regions (Fig. 5C). Our finding suggests that this spatial concentration of research in these regions may have led to a substantial overestimation of the global influence of eddies in aggregating forage fauna. When our analysis is restricted to regions with high mean eddy amplitude, the proportion of eddies affecting forage fauna increases substantially, reaching on average 35% (Fig. S8), compared to the mean 12% when the analysis is performed globally (Fig. 3). However, the forage fauna response is not systematically consistent with an oasis effect, with mixed positive and negative response. The uneven distribution of our dataset (Fig. 5A), which under samples regions with high eddy amplitudes where prior studies have been conducted (Fig. 5C), may also potentially lead to a slight overestimation of null-effect eddies in our analysis. Specifically, the publicly available acoustic database used in our study lacks observations in the northwest Pacific, the northwest Atlantic and the northern part of the Indian Ocean (Fig. 5A), where strong currents occur. For example, the northwest Atlantic, where four previous studies have reported increased forage fauna density in AE[2,12,13,45], is not covered by our dataset. Therefore, it is

possible that specific mechanisms, such as the bottom-up aggregation effect occur at a higher rate in these under-sampled regions, but the present dataset does not allow for such evaluation, underscoring the need for publicly available sonar data in these areas.

### Interpreting sonar signal across eddies

Sonars represent valuable and efficient tools for observing marine ecosystems and currently stand as the sole method for consistently investigating forage fauna across depths on a global scale[26,28,46]. Yet, like any observational approach, they come with inherent limitations that warrant consideration. Acoustic sampling devices, typically mounted on the bottom of vessels hulls, are unable to effectively sample the first 20 m of the water column due to contamination from bubbles and surface reflections. Additionally, our ability to interpret forage fauna species is constrained by the use of a single acoustic frequency (38 kHz)[26]. This limitation hinders our ability to differentiate responses to the eddy trapping effect[12,45], as different forage fauna species have varying swimming ability. The 38 kHz signal primarily captures gas-bearing organisms like fish with swimbladders or siphonophores[29], limiting our ability to comprehensively identify forage fauna species. While changes in acoustic backscatter serves as an accepted measure of change in organism density[28], we recognize the

potential challenge in interpreting these changes due to shifts in forage fauna composition[47,48]. For the purposes of this analysis, we have assumed that acoustic backscatter signals principally reflect changes in density rather than composition. Constructing datasets similar to the 38 kHz used here, incorporating additional acoustic frequencies and/or employing broad-band echosounder hold promise for enhancing our understanding of changes in the forage fauna community inside and outside the eddies. Furthermore, the integration of global datasets encompassing midwater trawl data alongside acoustic data could significantly contribute to our understanding of eddy effects.

In summary, we have integrated a comprehensive, high-resolution global dataset of sonar-detected forage fauna across the upper 20–750 m of the ocean with a global atlas of oceanic eddy trajectories. Leveraging these extensive datasets, we assessed the influence of oceanic mesoscale eddies on forage fauna, a key trophic component in the pelagic ecosystem, across diverse oceanic regions. Our findings reveal that the vast majority of eddies sampled by our dataset do not exert a significant influence on forage fauna density as detected by acoustic echosounders. Focusing on the few eddies significantly affecting forage fauna, eddy amplitude and water-mass trapping are the only eddy characteristics exerting a pivotal role on forage fauna density, suggesting a trapping effect on forage fauna rather than the typical bottom-up structuring hypothesis.

The limited influence of eddies on forage fauna observed in our dataset challenges the prevailing hypothesis of a widespread oasis or aggregating effect of eddies on forage fauna. It is important to note that our database primarily covers open-ocean quiescent waters, unlike previous research focused on strong current systems in the vicinity of the coast[2,13]. These earlier studies, relying on altimetry to eddy detection, primarily targeted strong eddies, which are most detectable using this method (Fig. 5). This may explain why previous research mainly reported an oases effect (Fig. 5). It is however noteworthy that, even when concentrating solely on the 5% strongest eddies in our dataset (50 eddies), 75% of them do not induce a significant forage fauna response. The complementarity of our results with those of prior studies suggests that the oases effect in the global ocean might be confined to the strongest eddies occurring mostly in high-energy current systems. However, in most of oceanic regions outside these conditions, oceanic would likely exert a marginal effect on forage fauna. To substantiate this hypothesis, future oceanographic surveys should systematically sample across regions with varying levels of eddy activity.

Our results also suggest that beyond the abundance of forage fauna, other mechanisms may contribute to the aggregation of top predators in eddies. Physical characteristics such as deeper thermoclines, typically found in Anticyclonic Eddies (AE)[35] characterized by warmer waters, may provide a wider thermal niche for predators. This wider range could enable predators to remain at deeper depths for longer periods, enabling facilitating access to deeper prey that would otherwise be inaccessible[9]. To better grasp the mechanisms underlying the enhanced abundance of top predators in eddies, future studies should delve into additional eddy characteristics such as amplitude and eddy surrounding environmental dynamics. Integrating these factors into their analyses would provide a deeper understanding of the underlying mechanisms. Expanding the scope of datasets to include regions with strong eddy amplitudes, as demonstrated in our study, would illuminate the variety of biological responses to eddies on a global scale. This underscores the importance of large-scale oceanic expeditions and initiatives facilitating opportunistic data collection, thus enabling the sampling of a broader range of eddies and the construction of comprehensive global datasets in currently under-sampled regions. Furthermore, direct observations of prey-predator interactions within eddies, coupled with continuous monitoring of eddy dynamics and biology using automated instruments like sailing drones[20] or profiling floats with echosounders,

would significantly enrich our comprehension of all marine life components within oceanic eddies.

## Methods

### Eddy database
The eddy database was used from the altimetric Mesoscale Eddy Trajectories Atlas (META3.2 DT), produced by SSALTO/DUACS and distributed by AVISO+ (https://aviso.altimetry.fr) with support from CNES, in collaboration with IMEDEA (product META3.2 EXP DT)[27].

The eddies are detected and tracked on absolute dynamic topography (ADT) satellite images[49] on a daily basis. The database provides the exact eddy shapes with the position of the centre, the speed edge contours (the eddy region where the water spins the quickest), the effective edge contours (the extreme eddy contour based on ADT). The eddy type (cyclonic and anticyclonic), the amplitude (i.e. the ADT difference between the eddy edge and the centre, a metric for the eddy strength), the effective area (i.e. the surface included in the effective contour), the effective radius (i.e. the mean distance between the eddy centre and the effective border) are also available for each eddy. The ability of each eddy to track water masses (i.e. the non-linearity)[1] was calculated as the ratio between the spatial distance covers by the eddy and the rotational speed of the eddy. This was done on a 5 days window to limit the errors of daily eddy detection.

Eddies with a lifespan superior or equal to 14 days were kept. Eddies are available for the 1993–2021 period, and present everywhere around the world, with fewer eddies along the equator. We sampled eddies with an amplitude ranging from 0.004 to 0.698 m, an effective area from 2464 to 107,984 km$^2$ and lifespan from 14 days to 1000 days (Fig. S9).

### Acoustic data
Georeferenced single-beam acoustic data at 38 kHz, spanning from 2001 to 2020, was collated from public databases, sourced by Australian, British, French, Peruvian and Spanish research programs in the Pacific, Atlantic and Indian Oceans (Table S2). We limited all acoustic profiles to the common largest depth range available in the dataset, from 20 to 750 m depth, and they were interpolated to a consistent vertical resolution of 10 m[26]. Profiles where the sun was between 0° and 18° below the horizon were removed to avoid vertical migration events during dawn and dusk, following the astronomical definition of twilight. Profiles where the seabed was above 1000 m depth were removed to exclude continental shelves from the analysis. The Nautical Area Scattering Coefficient (NASC, m$^2$/nmi$^2$), a proxy of forage fauna density[28], was used. In the total database, 118,813 vertical profiles were available and distributed in all oceans (Fig. S1A).

### Overlapping
For each acoustic observation, the spatially closest eddy was attributed based on the distance between the acoustic data position and the eddy center position. The exact effective contour of each eddy was then used to determine if the acoustic observation was inside or outside the eddy (Fig. 1D, grey polygon). For some detailed analysis, when the acoustic observation was located inside the eddy, an eddy region was associated: the effective border (which corresponds to 0–30% of the distance between the effective border and the center), the speed border (same with the speed border and the center), the core, and the intern (everything between the core region and the border region). The core/intern/borders regions split was done based on previous publications which demonstrated that sometimes the main signal occurs only in the eddy core (dominant upwelling/downwelling effect), and/or at the eddy border (where small scale processes, fronts, are stronger)[13,21,37]. Finally, a control (i.e. outside) band was defined as the spatial ribbon outside the eddy between the effective border and two times the mean eddy radius, and sampled during the same

night/day period. Only eddies with acoustic data in at least 2 sampled regions and the control region are retained in the study.

## Anomalies

For each eddy, the NASC mean vertical profiles were calculated inside the eddy and outside the eddy (Fig. 1). Then, vertical profiles of NASC anomalies were calculated at the scale of each eddy as the mean inside vertical profile minus the mean outside vertical profile, divided by the mean outside vertical profile. Only coherent day-time and night-time periods were used for the control vertical profile as the inside vertical profile (i.e. day period was compared to day, and night period to night). In a last step, the NASC anomaly vertical profiles of each eddy were averaged among all sampled eddies by eddy type.

Surface chlorophyll-*a* was extracted from GLOBCOLOUR[50] at a daily resolution on a 4 km spatial scale along the track of acoustic cruises. SST was extracted from the ESA SST dataset[31] at a daily resolution on a 5 km spatial scale along the track of acoustic cruises too. In the same way, chlorophyll and SST anomalies of each eddy were calculated as ratio (inside−outside)/outside values.

## Statistical analysis

For each sampled eddy, the distribution of the mean inside NASC values were compared to the distribution of the mean outside NASC values with a Wilcoxon test[51], by vertical layer: 0–200 m (epipelagic layer) and 200–750 m (mesopelagic layer) (Fig. S3). The possible results of the Wilcoxon tests were:

(1) "Null": none significant difference the inside and outside NASC distributions (same forage fauna density inside the eddy compared to the outside);

(2) "Decrease": significantly lower inside NASC distribution than the outside (a decrease of the forage fauna density inside the eddy compared to surrounding waters);

(3) "Increase": significantly higher NASC inside distribution than the outside (an increase of the forage fauna density inside the eddy compared to surrounding waters).

The number of null, increasing and decreasing eddies was summed across the whole dataset and represented in terms of percentage for the two vertical layers (0–200 m and 200–750 m) and the two eddy types (CE and AE).

In the same way, inside and outside SST and chlorophyll-a concentration values were compared based on Wilcoxon test for each sampled eddy, and then summed across the whole dataset.

All the analysis and the Figures were done on the R software (version 4.2.1), with custom made codes[52]. For the maps, the coastlines of the continents were extracted from the *rnaturalearth* package[53].

## Reporting summary

Further information on research design is available in the Nature Portfolio Reporting Summary linked to this article.

## Data availability

All data used in the present study are publicly available, excepted three acoustic surveys. Eddy trajectories and characteristics are available on the AVISO data page (https://www.aviso.altimetry.fr/en/data/products/value-added-products/global-mesoscale-eddy-trajectory-product.html)[27]. Acoustic data can be accessed through diverse internet repositories indicated in Supplementary Table S2. Raw acoustic data from the Malaspina circumnavigation expedition were processed using the open-source software Matecho v.6.7 following the standard procedures detailed in[26]. The rest of the acoustic repositories were already available as processed data. The three Mozambique channel surveys are not publicly available but are available upon request (anne.lebourges.dhaussy@ird.fr). SST and chlorophyll data were downloaded through Copernicus website. The products ID are:

"SST_GLO_SST_L4_REP_OBSERVATIONS_010_024" (SST)[31] and "OCEANCOLOUR_GLO_BGC_L4_MY_009_104" (chlorophyll)[32]. Source data are provided with this paper. A sample of the data are available on the public GitHub deposit[52]. Source data are provided with this paper.

## Code availability

Data analysis was conducted with custom-made analysis routines in R. All codes are available on the following GitHub deposit: https://github.com/auroreRECE/eddy_micronecton/tree/main[52].

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

## Acknowledgements

We acknowledge the Australian Integrated Marine Observing System (IMOS), the French National Research Institute for Sustainable Development (IRD), the British Antarctic Survey (BAS), the Peruvian Marine Institute (IMARPE), the Pierre and Marie Curie University (UPMC) and the Spanish National Research Council (CSIC) for their generous and invaluable contributions to the public acoustic databases used in the present study. We acknowledge Geir Pedersen for his help, and the cruise member of the One Ocean Expedition. Financial support was provided by the European Union "Pacific-European-Union-Marine Partnership" programme (Agreement FED/2018/397-941). This publication was produced with the financial support of the European Union and Sweden. Its contents are the sole responsibility of the authors and do not necessarily reflect the views of the European Union and Sweden. Financial support was provided by the New Zealand Government "Climate Science for Ensuring Pacific Tuna Access" project (Work Package Number: WPG-0103602). Aurore Receveur is part of the MAESTRO group co-funded by the Centre for the Synthesis and Analysis of

Biodiversity (CESAB) of the Foundation for Research on Biodiversity (FRB), and by France Filière Pêche.

## Author contributions

Conceptualization: AR, CM, PL, VA, SN. Methodology: AR, AA, ML, AB, CM, SN. Investigation: AR, ML, AB, CM, CD. Funding acquisition: SN, CM. Writing—original draft: AR, ML. Writing—review and editing: AR, CM, ML, AA, CD, AB, SC, VA, LB, ALD, PL, SN.

## Competing interests

The authors declare no competing interests.
