## [Peer Review File · Nature Communications]

A rare oasis effect for forage fauna in oceanic eddies at the
global scaleEditorial Note: This manuscript has been previously reviewed at another journal that is not operating a transparent peer review scheme. This document only contains reviewer comments and rebuttal letters for versions considered at *Nature Communications*. Mentions of the other journal have been redacted.

REVIEWER COMMENTS

Reviewer #1 (Remarks to the Author):

I appreciated the opportunity to re-review a “Life oases in oceanic eddies are more exception than the rule.” As mentioned previous, I was impressed with the dataset and overall amalgamation analyses undertaken by the authors – it truly is a breadth of data. I still feel that the manuscript reads largely as a rebuttal to a previous eddies as ocean hotspots papers rather than the strength I am glad to see the authors include more explicitly, of introducing new theory or hypotheses to the literature. Sampling differences (on behalf of the previous paper and this paper) might be the main difference in findings rather than a “paradigm shift” as the authors state. I think it is a really important finding that many if not most eddies lack biological relevance to the mesopelagic community but I would like to see the authors take this same tact of figuring out what processes and patterns are found in their data and analyses presented while also presenting the limitations of their study (similar to the acoustic limitations in the discussion). Again, this is improved compared to the previous version but I want to point out this logical trap in the abstract again:

“However, limited sampling of these eddies in past research has restricted the generalization of this aggregation hypothesis for the forage fauna. This study addresses this limitation by examining the response of forage fauna detected by shipboard acoustics by a record-breaking 999 eddies across the oceans.”

Since the study areas between this paper and the previous eddy paper that this is rebutting are different, I think the authors need to be more realistic here. More eddies does not necessarily mean that this study is doing a better job capturing the oceanic food web.

Instead I think this paper is showing that many eddies and maybe the majority simply do not have the same effect as previously observed. Figure 5 and the discussion is hugely helpful for addressing this.

I have some more detailed comments following:

L68-69: “In contrast, acoustic echosounder data offer high-resolution insights into the vertical structure of forage fauna when crossing eddies^{15,17}” – You provided a caveat on net tows in the previous sentence but none is provided here for acoustics (e.g. no species differentiation).

L193-195: “Despite the evident diversity in SST and chlorophyll responses observed in both CE and AE, our results underscore significant anomalies, signifying that the majority of sampled eddies in this study exhibit distinct physical and biogeochemical signature.” – This sentence is not clear as writing. Can you please simplify to the result?

L200-201: I’ve stated this previously but see the same issue here: “Strikingly, contrary to previous studies based on limited observations ^{2,13}” – this sets up a dichotomy that I simply do not see existing. The two studies looked at different regions largely so adding more eddies regionally will not solve this problem. I think the result is still hugely interesting in that most eddies do not have an “oasis effect.” I would suggest you lead with that, discuss why your findings are such, and then state the difference with previous studies rather than pinning your study to these previous findings.

L225: “To deepen the analysis” -> “To extend the analysis”

L282-283: I would replace “only” with “primarily” or “at a higher rate”

Reviewer #2 (Remarks to the Author):

I previously reviewed this manuscript in a submission to [journal name redacted] and agreed to review this revised submission. I still think this is an extremely valuable

contribution to the literature and should absolutely be published. However, in my opinion, while the scale of the analysis (and sheer work required to deal with these data) is commendable and on par with the “big” data analyses often published in high impact journals, I believe this manuscript still has some fundamental flaws that prevent me from recommending it for publication at Nature Communications.

One of the previous reviewers commented:

I think it is possible that previous studies focused on eddy-regions where these ecological effects are greatest and by diversifying your sampling, you are including less biologically important eddies. There are also some well know eddy systems in the Pacific (e.g. Gulf of Alaska, Papagayo) that are not sampled and have longer ages that could potentially support the bottom up hypothesis. This puts the authors at risk of the same fault of previous studies, in generalizing beyond what the data are able to tell you. L267-269 makes this point explicitly “On the basis of our results, it is strongly recommended that the next studies investigating the eddy influence on marine life first carefully examine the eddy characteristics and use them in their analysis to go deeper in the understanding of the mechanisms.” I would like to see the authors take this same tact of figuring out what processes and patterns are found in their data and analyses presented while also presenting the limitations of their study (similar to the acoustic limitations in the discussion).

While the revision does more explicitly treat the limitations of the current study, I still believe the fundamental flaw is in the fact that most of the eddies treated here are in specific, largely quiescent regions of the global ocean. Little to no data is included from the energetic western boundary currents from which most eddies are born and absolutely where the highest amplitude eddies are found. For example, in comparing Fig 1 from Yang et al 2016 (as one of many examples of the ocean energy regimes; <https://agupubs.onlinelibrary.wiley.com/doi/full/10.1002/2015JC011513>) with Fig 1 from the current submission, it is clear how biased the sampling of eddies is relative to these boundary currents. This discrepancy is also clear in the current Fig 5 which nicely shows the distribution of other studies that have asked similar questions and, notably, most that have found evidence to support the various biological enhancement mechanisms are indeed located in higher energy regions that are largely distinct from the areas considered in this

paper. That is not to say this contribution is not useful, rather I think the sweeping statements about the impact of eddies on biology are well beyond what they should be given the restricted spatial coverage (which I argue fundamentally skews these results, conclusions and interpretation). My recommendation is to scale the language to better match the conclusions, which largely consider low amplitude eddies in gyre-like, quiescent ocean regions. This is one of the few things that all 3 reviewers commented on previously that I don't think was adequately addressed in the revision.

Other comments:

Language and writing style throughout makes the manuscript somewhat difficult to read. Suggest significant attention in the revision should be given to flow, clarity and grammar which would significantly improve the read-ability of the paper.

Reviewer #1:

I appreciated the opportunity to re-review a “Life oases in oceanic eddies are more exception than the rule.” As mentioned previous, I was impressed with the dataset and overall amalgamation analyses undertaken by the authors – it truly is a breadth of data. I still feel that the manuscript reads largely as a rebuttal to a previous eddies as ocean hotspots papers rather than the strength I am glad to see the authors include more explicitly, of introducing new theory or hypotheses to the literature. Sampling differences (on behalf of the previous paper and this paper) might be the main difference in findings rather than a “paradigm shift” as the authors state. I think it is a really important finding that many if not most eddies lack biological relevance to the mesopelagic community but I would like to see the authors take this same tact of figuring out what processes and patterns are found in their data and analyses presented while also presenting the limitations of their study (similar to the acoustic limitations in the discussion). Again, this is improved compared to the previous version but I want to point out this logical trap in the abstract again: “However, limited sampling of these eddies in past research has restricted the generalization of this aggregation hypothesis for the forage fauna. This study addresses this limitation by examining the response of forage fauna detected by shipboard acoustics by a record-breaking 999 eddies across the oceans.”

Since the study areas between this paper and the previous eddy paper that this is rebutting are different, I think the authors need to be more realistic here. More eddies does not necessarily mean that this study is doing a better job capturing the oceanic food web. Instead I think this paper is showing that many eddies and maybe the majority simply do not have the same effect as previously observed. Figure 5 and the discussion is hugely helpful for addressing this.

We thank the reviewer for these comments on our manuscript. We have removed this sentence in the abstract, and it is now as follows: ‘Previous studies have posited that marine predators are drawn to these eddies due to an aggregation of their forage fauna. In this study, we examine the response of forage fauna, detected by shipboard acoustics, across a broad sample of a thousand of eddies across the world’s oceans.’

I have some more detailed comments following:

L68-69: “In contrast, acoustic echosounder data offer high-resolution insights into the vertical structure of forage fauna when crossing eddies^{15,17}” – You provided a caveat on net tows in the previous sentence but none is provided here for acoustics (e.g. no species differentiation).

We agree with the reviewer that both tools have pro and cos. We have added it for the echosounder too: ‘While net tows are effective in species examination, they struggle to adequately resolve mesoscale structures without dense, repeated sampling across multiple depth ranges and surveys. On the other hand, although acoustic echosounder data lack the ability to identify the species without specific validation, they offer high-resolution insights into the vertical structure of forage fauna when crossing eddies^{15,17}.’

L193-195: “Despite the evident diversity in SST and chlorophyll responses observed in both CE and AE, our results underscore significant anomalies, signifying that the majority of sampled eddies in this study exhibit distinct physical and biogeochemical signature.” – This sentence is not clear as writing. Can you please simplify to the result?

We are sorry for this lack of clarity. We wanted to say that we found various results for the SST and the chlorophyll in eddies: increased and decreased of SST and chlorophyll occur both in cyclonic and anticyclonic eddies, and not only decrease of SST in cyclonic eddies for example.

But, despite this variety, the percentage of eddies with a significant signal in SST and chloro remains high, result that go in the same way as the literature, and therefore validate our methodology. We cut the sentence in two parts to clarify: ‘Our investigation revealed significant anomalies in both positive and negative directions for SST and chlorophyll both in CE and AE, consistent with findings in previous literature^{34,35,36,38,39}, thus confirming the robustness of our methodology.’.

L200-201: I’ve stated this previously but see the same issue here: “Strikingly, contrary to previous studies based on limited observations 2,13” – this sets up a dichotomy that I simply do not see existing. The two studies looked at different regions largely so adding more eddies regionally will not solve this problem. I think the result is still hugely interesting in that most eddies do not have an “oasis effect.” I would suggest you lead with that, discuss why your findings are such, and then state the difference with previous studies rather than pinning your study to these previous findings.

We removed the comparison with the previous publications in this paragraph and staid focus on our results: ‘Conversely, the examination of forage fauna reveals a notable absence of significant changes (Fig. 3) in most CE and AE (86% and 90%, respectively, showed no effect in the epipelagic layer, and in 84% and 88%, respectively, in the mesopelagic layer). In the epipelagic layer, a significant increase (i.e. an oasis effect) in forage fauna is found in only 5% and 7% of AE and CE, respectively. The most pronounced indication of an oasis effect is observed for AE in the mesopelagic layer, where 9% of eddies exhibit a significant increase in forage fauna, whereas CE show a slightly larger proportion (11%) of forage fauna decrease at these depths.’.

L225: “To deepen the analysis” -> “To extend the analysis”

Done.

L282-283: I would replace “only” with “primarily” or “at a higher rate”

We rewrote the sentence (‘Therefore, it is possible that specific mechanisms, such as the bottom-up aggregation effect occur at a higher rate in these under-sampled regions, but the present dataset does not allow for such evaluation, underscoring the need for publicly available sonar data in these areas.’).

Reviewer #2:

I previously reviewed this manuscript in a submission to Nature and agreed to review this revised submission. I still think this is an extremely valuable contribution to the literature and should absolutely be published. However, in my opinion, while the scale of the analysis (and sheer work required to deal with these data) is commendable and on par with the “big” data analyses often published in high impact journals, I believe this manuscript still has some fundamental flaws that prevent me from recommending it for publication at Nature Communications.

One of the previous reviewers commented:

I think it is possible that previous studies focused on eddy-regions where these ecological effects are greatest and by diversifying your sampling, you are including less biologically important eddies. There are also some well know eddy systems in the Pacific (e.g. Gulf of Alaska, Papagayo) that are not sampled and have longer ages that could potentially support the bottom up hypothesis. This puts the authors at risk of the same fault of previous studies, in generalizing beyond what the data are able to tell you. L267-269 makes this point explicitly “On the basis of our results, it is strongly recommended that the next studies investigating the eddy influence on marine life first carefully examine the eddy characteristics and use them in their analysis to go deeper in the understanding of the mechanisms.” I would like to see the authors take this same tact of figuring out what processes and patterns are found in their data and analyses presented while also presenting the limitations of their study (similar to the acoustic limitations in the discussion).

While the revision does more explicitly treat the limitations of the current study, I still believe the fundamental flaw is in the fact that most of the eddies treated here are in specific, largely quiescent regions of the global ocean. Little to no data is included from the energetic western boundary currents from which most eddies are born and absolutely where the highest amplitude eddies are found. For example, in comparing Fig 1 from Yang et al 2016 (as one of many examples of the ocean energy regimes; <https://agupubs.onlinelibrary.wiley.com/doi/full/10.1002/2015JC011513>) with Fig 1 from the current submission, it is clear how biased the sampling of eddies is relative to these boundary currents. This discrepancy is also clear in the current Fig 5 which nicely shows the distribution of other studies that have asked similar questions and, notably, most that have found evidence to support the various biological enhancement mechanisms are indeed located in higher energy regions that are largely distinct from the areas considered in this paper. That is not to say this contribution is not useful, rather I think the sweeping statements about the impact of eddies on biology are well beyond what they should be given the restricted spatial coverage (which I argue fundamentally skews these results, conclusions and interpretation). My recommendation is to scale the language to better match the conclusions, which largely consider low amplitude eddies in gyre-like, quiescent ocean regions. This is one of the few things that all 3 reviewers commented on previously that I don't think was adequately addressed in the revision.

We thank the reviewer for this feedback. We have nuanced our results, and we added a paragraph (‘Regional sensitivity’) dedicated to the regional results and limits :

“The overwhelming majority of oceanic eddies identified in our dataset exhibit no discernible impact on forage fauna density, with consistently fewer than 10% of eddies exhibiting what is known as the “oasis effect” (i.e. an aggregating effect, 5%, 7%, 9% and 6%; Fig. 3) for both AE and CE in both the epipelagic and mesopelagic layers. A closer examination of the regional response of forage fauna across Longhurst⁴³ biogeochemical provinces (Fig. S7) reveals that regional results for most sampled provinces align with the global analysis, with an

overwhelming proportion of “null-effect” eddies (from 55 to 100 % in the sampled provinces). Nevertheless, three specific provinces - the East African coast, the Sub-Antarctic water ring and the Antarctic - exhibit slightly higher proportions of influencing eddies compared to the global results, although the dominance of null-effect eddies persists [85 (AE) and 68 % (CE); 79 (AE) and 72 % (CE); and 66 (AE) and 63 % (CE) respectively]. Eddies with significant effects on forage fauna tend indeed to cluster in subtropical regions and southern latitudes, in high energetic current systems such as the South Subtropical Convergence province⁴⁴ and where eddies have strongest mean amplitudes (Fig. 5A, 5B). Notably, previous studies^{13,41,45} highlighting a significant effect of eddies on forage fauna, often reporting an increasing fauna effect in CE, were primarily located in these specific high-amplitude eddy regions (Fig. 5C). Our finding suggests that this spatial concentration of research in these regions may have led to a substantial overestimation of the global influence of eddies in aggregating forage fauna. When our analysis is restricted to regions with high mean eddy amplitude, the proportion of eddies affecting forage fauna increases substantially, reaching on average 35% (Fig. S8), compared to the mean 12 % when the analysis is performed globally (Fig. 3). However, the forage fauna response is not systematically consistent with an oasis effect, with mixed positive and negative response. The uneven distribution of our dataset (Fig. 5A), which under samples regions with high eddy amplitudes where prior studies have been conducted (Fig. 5C), may also potentially lead to a slight overestimation of null-effect eddies in our analysis. Specifically, the publicly available acoustic database used in our study lacks observations in the northwest Pacific, the northwest Atlantic and the northern part of the Indian Ocean (Fig. 5A), where strong currents occur. For example, the northwest Atlantic, where four previous studies have reported increased forage fauna density in AE^{2,12,13,45}, is not covered by our dataset. Therefore, it is possible that specific mechanisms, such as the bottom-up aggregation effect occur at a higher rate in these under-sampled regions, but the present dataset does not allow for such evaluation, underscoring the need for publicly available sonar data in these areas.” ‘

We also changed the concluding remarks : ‘It is important to note that our database primarily covers open-ocean quiescent waters, unlike previous research focused on strong current systems in the vicinity of the coast^{2,13}. These earlier studies, relying on altimetry to eddy detection, primarily targeted strong eddies, which are most detectable using this method (Fig 5). This may explain why previous research mainly reported an oases effect (Fig. 5). It is however noteworthy that, even when concentrating solely on the 5% strongest eddies in our dataset (50 eddies), 75% of them do not induce a significant forage fauna response. The complementarity of our results with those of prior studies suggests that the oases effect in the global ocean might be confined to the strongest eddies occurring mostly in high-energy current systems. However, in most of oceanic regions outside these conditions, oceanic would likely exert a marginal effect on forage fauna. To substantiate this hypothesis, future oceanographic surveys should systematically sample across regions with varying levels of eddy activity.’,

We removed ‘the paradigm shift’ expression. We also removed the comparison with previous publications in the abstract.

Other comments:

Language and writing style throughout makes the manuscript somewhat difficult to read. Suggest significant attention in the revision should be given to flow, clarity and grammar which would significantly improve the read-ability of the paper.

We carefully read and correct the English and grammar in the new version of our manuscript.

REVIEWERS' COMMENTS

There are no reviewer comments from this round of review.